# Porphyrin-Based Molecules in the Fossil Record Shed Light on the Evolution of Life

**Juan D. Ayala** *, **Elena R. Schroeter** and **Mary H. Schweitzer** *

Department of Biological Sciences, North Carolina State University, Raleigh, NC 27695, USA; easchroe@ncsu.edu
* Correspondence: jdayala@ncsu.edu (J.D.A.); mhschwei@ncsu.edu (M.H.S.)

**Abstract:** The fossil record demonstrates the preservation of porphyrins (e.g., heme) in organic sediments and the fossilized remains of animals. These molecules are essential components in modern metabolic processes, such as electron transport (cytochromes) and oxygen transport (hemoglobin), and likely originated before the emergence of life. The integration and adaptation of porphyrins and structurally similar molecules (e.g., chlorophylls) are key aspects in the evolution of energy production (i.e., aerobic respiration and photosynthesis) and complex life (i.e., eukaryotes and multicellularity). Here, we discuss the evolution and functional diversity of heme-bound hemoglobin proteins in vertebrates, along with the preservation of these molecules in the fossil record. By elucidating the pivotal role of these molecules in the evolution of life, this review lays the groundwork necessary to explore hemoglobin as a means to investigate the paleobiology of extinct taxa, including non-avian dinosaurs.

**Keywords:** molecular paleontology; heme; porphyrin; hemoglobin; evolution; chemical evolution; preservation





## 1. Introduction

Porphyrin-containing proteins (e.g., cytochromes, hemoglobin, and myoglobin) have been essential components of metabolism since the earliest form of life emerged at least 3.4 billion years ago [1,2], and are required for the metabolism of virtually all taxa, from bacteria to plants to humans, in the modern era. This review will focus on heme–hemoglobin interactions. The oxygen-binding activities of heme porphyrin in particular make hemoglobin critical for respiration in most extant vertebrate taxa. Composed of four pyrrole ($C_4H_4NH$) rings connected via methine bridges ($R_1$-CH=$R_2$) (Figure 1), heme is responsible for the mediation of oxygen and carbon dioxide ($CO_2$) exchange throughout vertebrate tissues. Some plants also use hemoglobin—in the form of leghemoglobin—to facilitate the activity of anoxic nitrogen-fixing bacteria in symbiosis with plant tissues [3]. Hemoglobins contain a wealth of information on the physiology of taxa bearing them, their phylogenetic history, and their environments. Thus, the successful characterization of hemoglobin from extinct organisms could open an unprecedented window into their paleobiology unattainable from gross skeletal morphology, or even more abundant molecules (e.g., collagen).

Proteins and other biomolecules are assumed to be labile, and most begin to degrade shortly after an organism dies. However, molecular techniques, including paleoproteomics and various forms of spectroscopy (e.g., Raman Spectroscopy), suggest that small concentrations of endogenous organic material are preserved in fossil tissues [4–6]. Because their intrinsic structural properties greatly enhance stability, heme porphyrins and similar molecules (e.g., chlorophyll-$\alpha$) have been reported in plant-derived organic sediments (e.g., oil shales [7–9] and asphalts [9]) and fossil tissue of animals [10–14]. Peptide sequences of hemoglobin have also been reported from archaeological and Pleistocene taxa [6,15–21], as well as specimens from deep time (i.e., greater than 1 Ma in age) [10,22,23]. Further, immunohistology methods have demonstrated immunolocalization of hemoglobin in fossil samples of various ages [5,22,24–28].

**Figure 1.** Pyrrole and base porphyrin ring. (**a**) Pyrrole ($C_4H_4NH$) is a five-member nitrogen-containing ring that forms tetrapyrrole molecules. (**b**) Unsubstituted porphin tetrapyrrole ring with beta (β) and meso (m) positions labeled along the periphery of the ring. (**c**) Numeric labels of the 20 carbon molecules forming the porphyrin rings.

It has been hypothesized that hemes recovered from preserved animal taxa may have derived from hemoglobin [10,11,14], and therefore may serve as a proxy for inferring its presence in fossil tissues. However, confident identification of hemoglobin-derived heme must also rule out other potential sources of this porphyrin, such as bacterial and vertebrate cytochromes or peroxidases. Here, we review the structure and function of porphyrins and other tetrapyrrole molecules in the early metabolism of life and the evolution of hemoglobin (beginning with its earliest hypothesized precursors), and discuss the preservation potential of this molecule in the fossil record. We examine methods to confidently identify heme in fossil remains of vertebrates through the lens of its unique molecular structure and function. We start with an overview of the structure of porphyrins molecules followed by the conditions on the early Earth, under which life first evolved, and the role porphyrins and other tetrapyrroles played in the oxygenation of the planet and the evolution of aerobic life. We then delve into the function, structure, and diversity of vertebrate heme-bound proteins, with a specific focus on hemoglobin and its role in respiratory adaptations to environmental pressures. Next, we discuss the physical and chemical properties that contribute to the persistence of heme and/or hemoglobin in the fossil record, the diagenetic alterations that may occur at the molecular level, and how we may distinguish between heme derived from hemoglobin versus other heme-bound proteins. Finally, we explore the ways preserved hemoglobin may be used to investigate aspects of paleobiology that are currently inaccessible.

## 2. Porphyrins and Other Tetrapyrroles

### 2.1. Porphyrin Structure and Biological Function

Heme (Figure 2) belongs to a class of molecules called porphyrins, which are a type of tetrapyrrole. A pyrrole ($C_4H_4NH$) is a five-member ring that contains nitrogen (Figure 1a) [29,30]. When four of these rings are interconnected by methine bridges ($R_1$-CH=$R_2$) into a larger, macrocyclic ring, they form a tetrapyrrole (Figure 1b). These large, tetrapyrrole rings contain an alternating series of single and double bonds [29,30]. The electron pairs forming these bonds can shift freely across the molecule (i.e., they are delocalized), in such a way that connections between its constituent atoms oscillate between single and double-bond states [29,31,32]. This arrangement of atoms is known as an aromatic ring [29,31,32]. Because of the orientation of the nitrogen in its pyrrole sub-units, tetrapyrroles commonly form complexes with metal cations, such as iron (Fe), magnesium (Mg), copper (Cu), zinc (Zn), cobalt, (Co), and nickel (Ni), coordinated in the molecule's

core [30,33–35]. At the periphery of the ring, tetrapyrroles possess sites for potential substitutions [29,30]. In the heme tetrapyrrole, the core binds ferrous iron ($Fe^{2+}$) at its center, giving hemoglobin its familiar red color, and the substitutions at the periphery of the ring result in several distinct types of heme (e.g., heme *b*, and heme *c*) [29,33,36,37].

**Figure 2.** Heme *b*. The heme *b* variant is bound to the hydrophobic pocket of hemoglobin. Ferrous iron ($Fe^{2+}$) is bound to the center of the heme porphyrin through an ionic bond. Histidine residues of hemoglobin bind to the iron ion. Substitution along the porphin ring includes methyl ($-CH_3$) at C's 2, 7, 12, and 18, vinyl group ($-CH=CH_2$) at 3 and 8, and propionic acids ($-CH_2CH_2COOH$) at 3 and 17.

The aromatic nature of tetrapyrroles, along with their ability to bind metal ions, gives rise to diverse molecules with properties that are essential in many biological pathways. For example, heme is utilized for oxygen transport in hemoglobin [3,34,36], electron transport in cytochromes [34,38–40], and chlorophyll for light absorption in photosynthesis [36,41,42]. These features are also thought to contribute to the stability of its molecular structure [31]. In particular, the free-moving nature of the bonded electron pairs in aromatic tetrapyrroles makes them relatively stable molecules [31,32]. This is because the oscillation of their bonds between single and double makes all bonds essentially "one and a half", conferring greater stability than static single bonds [31]. In particular, the stability of heme is further heightened by the presence of the additional double bond compared to other types. This extra double bond is what distinguishes porphyrins such as heme from other biologically important tetrapyrroles such as chlorins (e.g., chlorophyll) and bacteriochlorin (a photosynthetic pigment found in non-oxygen-producing bacteria) (Figure 3) [32,43]. The aromatic nature of heme, along with its additional porphyrin double bond and its affinity to bind iron, are all stabilizing factors that are thought to increase its potential for preservation in deep time [33,44].

## 2.2. Role of Porphyrins in Early Life

Porphyrins are ubiquitous across modern taxa and are present in both prokaryotes and eukaryotes [34]. These molecules were likely present in the metabolism of organisms prior to the divergence of eukaryotes and prokaryotes, a hypothesis supported by the utilization of the same precursor molecules (5-aminolevulinic acid) in all observed biological synthesis pathways [34] and the presence of the intermediate molecule uroporphyrin in

all biologically active tetrapyrroles (e.g., heme porphyrins and chlorophyll chlorins) [7,36]. Because their incorporation into organisms is thought to have occurred extremely early in the development of life [45], it is necessary to go as far back as the origin of life itself to understand their biological significance and role in biological processes.

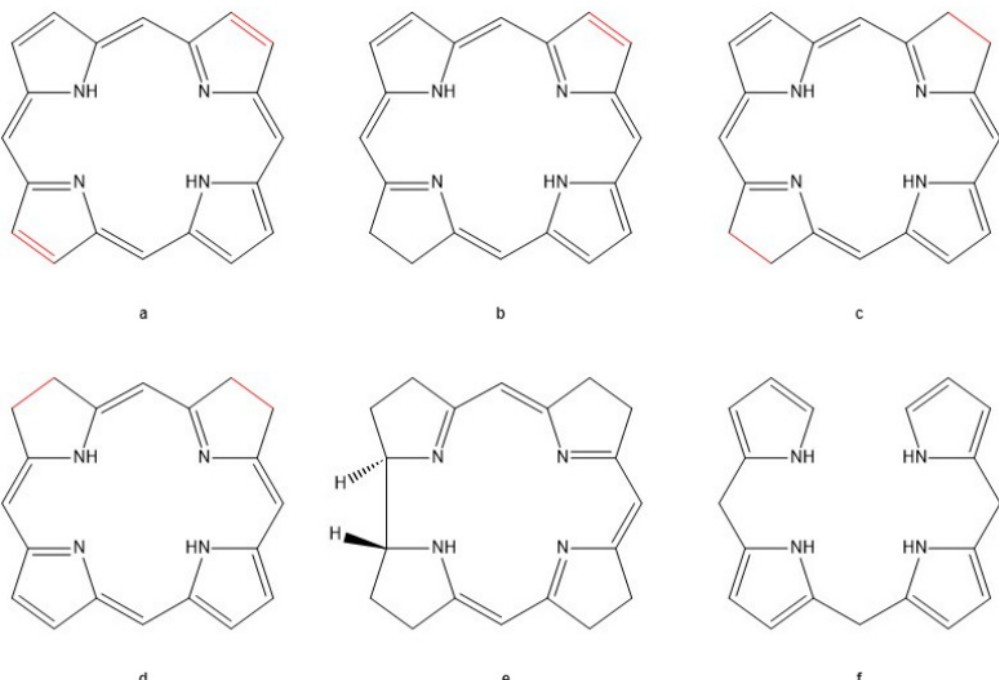

**Figure 3.** Tetrapyrrole pigments variants. (**a**) Porphin ring highlighting the two double bonds on opposite sides of the aromatic molecule. (**b**) Chlorin, the parent molecule for plant pigments like chlorophyll-$\alpha$. (**c**) Bacteriochlorin, the parent molecule of photosynthetic pigments in bacteria in anoxic photosynthesis. (**d**) Isobacteriochlorin, an isomer of bacteriochlorin. (**e**) Corrin, the parent pigment of vitamin B12. (**f**) Linear bilane, pigment molecules formed during the synthesis and degradation of heme molecules [35].

2.2.1. The Abiotic Formation of Porphyrin and Chemical Origin of Life

Following the formation of the earth, environmental conditions during the early Hadean era (4.6 Ga) were hostile and destructive to the formation of life. Greatly increased volcanic activity (compared to modern conditions) and frequent meteor impacts created an environment too hot for life to flourish [46,47]. However, as primordial oceans formed and the planet cooled during the Archean (4.0–2.5 Ga), conditions on Earth changed permitting the formation of prebiotic biopolymers (e.g., amino acids, carbohydrates, and RNA) from inorganic molecules such as $CO_2$ and ammonia [46,48,49]. There was little free oxygen present in the Earth's early atmosphere, which was rich in $CO_2$, nitrogen ($N_2$), and methane gases [34,36,47,48,50]. These reducing conditions led to an ocean rich in ferrous iron and sulfur [51–53]. Because free oxygen drives the rapid oxidation of most biomolecules, its presence in the ancient atmosphere would break apart the chemical bonds of prebiotic compounds as soon as they formed, creating an environment toxic to life [54,55]. Therefore, the anaerobic or dysaerobic (low free oxygen) and reducing conditions of the early Earth would favor the abiotic formation of biopolymers and their precursors [48,49], setting the stage for self-organizing molecules (i.e., RNA world hypothesis [56]) that would eventually form the earliest organisms [55]. Porphyrins were likely among the earliest prebiotic molecules to form and contributed to the development of life 3.8–3.4 Ga [2,56–60].

Like the early biopolymers that form the constituents of life even today, porphyrins may have formed abiotically [2,60–62], along the shores of primordial volcanic islands. The prebiotic oceans of the late Hadean and early Archean are hypothesized to retain a greater concentration of inorganic salts and amino acids compared to the modern oceans [62–64].

These amino acids were potentially generated abiotically via ultraviolet radiation [45,65,66]. The interaction between the salty ocean water containing various abiotic amino acids and volcanic eruptions provided the thermal and acidic conditions required for porphyrin synthesis, driving the conversion of amino acids into pyrroles [62–64]. Repeated cycles of heating and cooling condense pyrroles into porphyrins [63]. This abiotic synthesis has successfully generated porphyrins under conditions simulating that of early Earth [2,60–62].

Among these abiotically generated porphyrins was uroporphyrin, which is a possible prebiotic photosensitive compound that is also the universal porphyrin intermediate in all biologically relevant tetrapyrroles [2,36,67]. The global extreme heat provided by volcanic activity required for the abiotic generation of porphyrins in the Hadean and early Archean slowly gave way to less harsh conditions, making it unlikely that the porphyrins preserved in organic sediment and fossil tissue have an abiotic origin. However, the abiotically generated porphyrin co-factors may have been assimilated into the proto-metabolisms of early organisms [63].

The earliest organisms to evolve during the Archean were most likely anaerobic prokaryotes [53,59]. One of the earliest cofactors hypothesized to associate with anaerobic enzymes for energy transfer in such early life forms is iron–sulfur (Fe-S) clusters [59,66,68,69]. The reducing conditions of the Archean oceans would have permitted the spontaneous generation and environmental availability of Fe-S clusters [53,69]. Along with these Fe-S clusters, abiotically formed porphyrin molecules were also environmentally available and employed in early metabolic pathways, although it is uncertain when and how this association first occurred. It is hypothesized that early organisms must have developed ways to compensate for oxidative stress and/or light absorption in initial forms of photosynthesis, and porphyrins may have filled this role [2,61]. The different forms of tetrapyrroles, including porphyrins (heme), chlorins (chlorophylls), and bacteriochlorin, are all hypothesized to derive from uroporphyrins [36,41,66], and distinct forms of these molecules may have arisen from variations in redox potentials [42,66,70]. For example, heme *c* of cytochromes in the electron transport chains (ETC) possess a wider redox potential that is more conducive for electron transport than heme *b* [71].

### 2.2.2. This Distribution and Function of Porphyrins as Life Evolves

In our modern, aerobic world, porphyrins are incorporated into heme-containing cytochromes, which are essential components of the ETC in anaerobic [72–74] and aerobic respiration [40,73,75], as well as in photosynthesis [39,40]. In aerobic respiration, cytochromes pass electrons through the ETC, resulting in the reduction of oxygen into water by cytochrome oxidases [73,75,76]. Although oxygen is a crucial part of this process, molecular evidence suggests that cytochrome oxidases originated when Earth's environment was still anaerobic, or at least dysaerobic, and prior to the divergence of bacteria and archaea [1,72,75–78]. It is thought that cytochrome oxidases evolved from enzymes that reduce nitric oxide (NO) to $N_2$ and that aerobic respiration is derived from this process [72,75]. NO and small concentrations of free oxygen (created by the photolysis of water) present in the early Earth's atmosphere [48,75,79] would have had the potential to generate toxic reactive nitrogen species (RNS) and reactive oxygen species (ROS) [54,80], respectively. Ancestral cytochrome oxidase may have utilized NO as the terminal electron acceptor instead of oxygen when the atmosphere was still anaerobic and little free oxygen was present [75,76]. In this way, early cytochrome oxidases would serve as a defense against RNSs for early organisms. Accordingly, when the earth's atmosphere became oxygenated (see below), it is thought cytochrome oxidases transitioned to using oxygen as their final electron acceptor, as oxygen is more electronegative and therefore more efficient in this role [73,75]. The reduction of oxygen by cytochrome oxidases into water is thought to be the basis from which aerobic respiration was derived [76]. This change, then, would have allowed cytochrome oxidases to protect early aerobic life from ROSs in the same way ancestral forms (i.e., NO-reducing enzymes) guarded against RNSs [76]. Early organisms

with ETC competent to the utilization of oxygen may have had a selective advantage once atmospheric oxygen levels increased during the Great Oxygenation Event (GOE) [69,76].

The dispersal of aerobic metabolism in early organisms was likely facilitated by the transition from a reduced, anaerobic environment to an oxidized, aerobic environment [76,81]. This transition in the environment was likely driven by photosynthesis, a process contingent on tetrapyrrole molecules, bacteriochlorin, and chlorins. Photosynthesis is essentially the chemical reaction opposite of aerobic respiration; instead of breaking down organic molecules, photosynthesis captures solar energy through the implementation of bacteriochlorin and chlorin tetrapyrrole pigments to produce organic molecules [42,65,82,83].

### 2.2.3. The Origin of Photosynthesis and the Oxygenation of the Earth

The first photosynthetic organisms were likely anaerobic prokaryotes utilizing protein complexes that were unable to generate oxygen [42]. Modern anoxygenic photosystems utilize bacteriochlorophylls to capture light and have redox potential sufficient to oxidize electron donors such as hydrogen gas ($H_2$), $Fe^{2+}$, hydrogen sulfide ($H_2S$), formate, or oxalate [70], and it is hypothesized that early forms may have been similar [42,70]. In particular, graphite carbon isotopes from sediments 3.8 Ga provide evidence of early anoxygenic photosynthesis utilizing hydrogen as its electron donor [83]. Conversely, oxygenic photosynthesis, which oxidizes water molecules into molecular oxygen, requires a wider redox potential than possessed by bacteriochlorophylls [42,70] and is thought to have arisen through the development of chlorophyll-$\alpha$ (i.e., a chlorin tetrapyrrole) [70]. The evolution of chlorophyll, and thus oxygenic photosynthesis, in the ancestors of cyanobacteria and modern plants is hypothesized to be the main driver of the oxygenation of the environment during the GOE [42,47,81,84].

The GOE altered the redox chemistry of the environment on a global scale, transforming the planet to an oxidative state [68]. Evidence of the oxygenation of the atmosphere includes banded-iron formations (BIFS) [47,68,85] and mass-independent sulfur isotopes [68,86]. In the presence of oxygen, iron is converted from its ferrous state, $Fe^{2+}$, to its water-insoluble ferric state, $Fe^{3+}$. These ferric iron oxides then precipitated out of the water column to form sedimentary structures alternating in iron oxide-rich and iron-poor chert sediments [68,85], giving these structures a banded appearance. It is hypothesized that once ferrous iron was depleted from the oceans, the generation of ferric iron oxides subsided, consistent with the disappearance of banded-iron formations from the rock record (~2.3 Ga). With the oceans saturated with oxygen, free oxygen would have then been able to diffuse into the atmosphere. Additionally, there was a shift in sulfur isotopic signatures that coincided with the disappearance of BIFs between 2.3 to 2.4 Ga. Preservation of mass-independent sulfur isotope signal occurs in the rock record before the GOE. Free oxygen accumulation in ancient atmosphere, resulting in a shift to a mass-dependent sulfur isotopes, and greatly increased oxidative continental weathering [86].

### 2.2.4. Early Organismal Compensation for Oxygen Toxicity and the Evolution of Multicellularity

The accumulation of dissolved and atmospheric oxygen during the GOE was a critical transition to the evolution of eukaryotic, and, eventually, multicellular, life [53,58,81]. Oxygen functions as the final electron acceptor in the aerobic respiration pathways of prokaryotes and eukaryotes [87]. However, oxygen poses a hazard to both anaerobic and aerobic organisms because it generates ROSs, including superoxide and hydroxyl ions [54,55]. Oxygen and its ROS byproducts are highly reactive molecules that can interact with proteins, lipids, and nucleic acids, disrupting enzymatic processes, destroying lipid membranes, forming cross-linkages that disable proteins [88], and damaging DNA [54,89,90]. Ancient anaerobes likely exhibited metabolisms relatively similar to their modern counterparts [45], which are rich in Fe-S cluster proteins [69]. In the presence of oxygen species, these Fe-S clusters become irreversibly denatured or destabilized [69]. Nitrogenase, an enzyme responsible for the fixation of dissolved atmospheric $N_2$ into ammonia for amino acid synthesis in

bacteria and archaea, utilized Fe-S clusters as a cofactor, making it vulnerable in oxygenated environments [91]. Additionally, oxygenation limits the bioavailability of soluble $Fe^{2+}$, the most widely used metal cofactor in proteins, by oxidizing and precipitating iron [69]. The greatly increased oxidative stress generated by environmental oxygenation required anaerobic organisms to develop mitigation strategies to cope with this change. These may have included escape to low-oxygen environments (i.e., deep-sea hydrothermal vents [55]), development of tolerance, or the evolution of antioxidant defenses [55,69,70,87], such as heme-containing proteins [3].

In aerobic organisms, damage caused by ROSs generated from the environment or through metabolic processes is compensated for by peroxidase hemoproteins [57,92–94]. Coupled with cytochromes to generate energy gradients capable of producing 16–18 times more energy (i.e., adenosine triphosphate, ATP) compared to anaerobes [57], hemoproteins are critical to both the efficiency of aerobic respiration and as a guard against oxidative toxicity. Thus, the interaction between hemoproteins and the increased availability of oxygen after the GOE may have driven the dispersal of aerobic respiration. The widespread incorporation of aerobic respiration, in turn, allowed the evolution of more complex life forms [57,95]. For example, the energy surplus provided by aerobic respiration may have been favorable to the incorporation of the aerobic ancestor of modern mitochondria by anaerobic archaea, the endosymbiotic relationship of which is hypothesized to have given rise to eukaryotes 2 Ga ago [57,76,96,97]. Increases in energy output may have also permitted the formation of more complex biomolecules, such as steroids and fatty acids, giving rise to internal lipid membranes [57]. Lipid membranes, in turn, would have allowed the compartmentalization of eukaryotic cells (i.e., formation of organelles) [57], permitting greater control and complexity of signaling pathways within cells [57]. Ultimately, the energetic efficiency of aerobic respiration may have made cooperation and resource sharing between organisms viable, leading to the evolution of multicellular life during the Ediacaran (~600 Ma) [57,95].

With increased energy availability to drive proton pumps and other cellular mechanisms, multicellular life was favored. The emergence of multicellular organisms meant life was no longer restricted to isolated cells. Organisms could achieve greater masses and invade previously unviable environments. But, the development of tissues necessitated a means to transport materials from the organism's exterior to internal cells no longer in direct contact with the external environment. Aerobic respiration, in particular, would require a means to transport oxygen to distal and/or internal cells once passive diffusion was not sufficient to meet energetic demands. Globin hemoproteins may have provided a mechanism for such capture and transport of oxygen [3].

### 2.3. The Diversification of Globin Proteins, and Divergences within Vertebrata

The globin superfamily of hemoproteins is a wide and diverse group of proteins. They are typically composed of six to eight alpha-helical domains and are found in both prokaryotic and eukaryotic organisms [3,80,98,99]. Across taxa, globin proteins vary in structure and function, but all possess iron-heme porphyrins that facilitate the reversible binding of oxygen molecules [100]. Vertebrates express seven globin proteins, including myoglobin (Figure 4a) and hemoglobin (Figure 4b), which are tissue-specific proteins expressed in the muscle and blood tissue, respectively [101,102]. An ancestral globin is suspected to have existed before the divergence of prokaryotes and eukaryotes, derived from the hemoproteins used in redox metabolism (i.e., cytochromes) or to mitigate oxygen toxicity (peroxidases) [3,98,100,103].

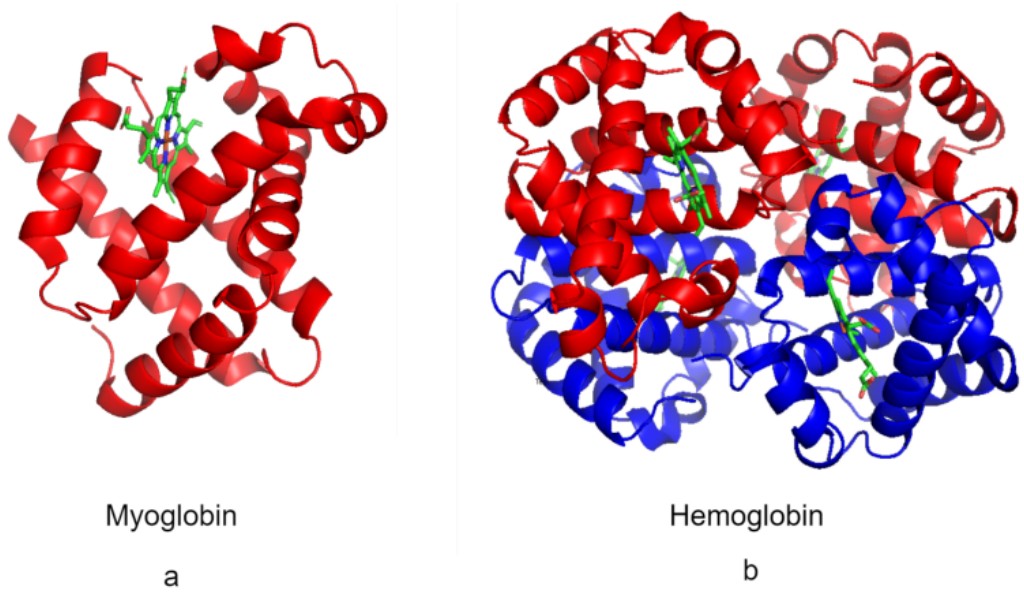

**Figure 4.** Myoglobin and hemoglobin. (**a**) Structure of myoglobin (Protein Data Bank ID: 3RGK) monomer (red) [104] and the heme cofactor (green). (**b**) The structure of the hemoglobin (Protein Data Bank ID: 1A3N) tetramer is composed of α subunits (red) and β subunits (blue) and the heme cofactor (green) [104].

Outside of vertebrata, organisms possess a variety of globin and "hemoglobin" proteins that are expressed with diverse functions and structures (see [99] and references therein). For example, flavohemoglobin is a chimeric globin protein expressed in bacteria and yeast. Composed of a heme domain and a flavin (a non-tetrapyrrole pigment) domain, they are utilized in denitrification and redox biochemical processes [80,99,105]. Leghemoglobin, found in legume plants (e.g., peanuts) functions as an oxygen scavenger in the roots of plants, allowing for nitrogen fixation by symbiotic anaerobic microbes [98,99]. Interestingly, based on gene sequencing data, plant globin is hypothesized to share ancestry with animal hemoglobin [98,99]; thus, this molecule originated before the divergence of plants and animals in deep time, about 1.6 Ga ago [99]. Neuroglobins (nerve globin) function in oxygen sensing and were likely present in the common ancestor of invertebrates and vertebrates prior to the divergence of these groups during the Ediacaran [106].

Within vertebrata, there is a divergence in globin structure between Agnatha (jawless vertebrates) and Gnathostomata (jawed vertebrates), lineages that diverged 510 Ma [107]. Agnathans possess globin with either one or two domains, and gnathostomes possess a four-domain (tetrameric) hemoglobin (Figure 4). The tetrameric molecules of hemoglobin in jawed vertebrates permit greater efficiency in oxygen transport compared to jawless vertebrates (e.g., lampreys) (see below) [108].

## 3. Structure and Function of Vertebrate Hemoglobin

In the red blood cells (RBCs) of jawed vertebrates, hemoglobin is a tetramer composed of four globin protein subunits (i.e., domains) that possess heme cofactors and are structurally similar but distinguishable by differing peptide sequences [102,109]. The most common composition of adult vertebrate hemoglobin is a combination of two α-like and two β-like subunits [102,108–110]. Each α and β subunit join together or dimerize to form an αβ protein complex or heterodimer, which then binds to another αβ-heterodimer to form a $\alpha_2 \beta_2$ tetramer [111]. Some vertebrates have alternate compositions that incorporate different subunits, such as the β/δ hybrid subunits of paenungulate clade (i.e., elephants, hyraxes, and manatees) [110,112,113]. Subunit expression may also vary depending on the stages of development (e.g., human fetal hemoglobin $\alpha_2\gamma_2$) [114].

Within a highly conserved, hydrophobic pocket of each of the four subunits resides an iron-containing heme prosthetic group, or cofactor. The heme is anchored in this domain

pocket by the interaction of its constituent iron with a histidine amino acid residue [111,115]. This heme-bound pocket is the binding site for oxygen or $CO_2$ during respiration [109].

### 3.1. The Physiological Role of Hemoglobin and Its Limitations

The binding of oxygen induces a conformational change (i.e., changes the molecular shape in response to the environment) that alters the oxygen-binding affinity of the hemoglobin subunit. In the deoxygenated 'relaxed' state (R-state), hemoglobin possesses a high oxygen-binding affinity. Once oxygen is bound to a subunit, a conformational change occurs that transitions the bound unit into a low oxygen-binding affinity 'tense' state (T-state). Across the hemoglobin molecule, the individual subunits experience a phenomenon known as cooperative binding, in which the loading of oxygen in one subunit induces a conformational change that increases the oxygen-binding affinity of the remaining, unloaded subunits [108,115,116] until all reach the T-state.

The reversible binding of oxygen to hemoglobin is facilitated by the difference in oxygen partial pressure ($PO_2$) between respiratory tissues (gills, skin, and lungs), RBCs, and distal body tissue (Figure 5) [117]. At the interface between respiratory tissue and RBCs, waste $CO_2$ is released from the body, and oxygen is loaded. RBCs then transport oxygen to the distal, reduced $PO_2$ tissues, unloading it for cytochrome-mediated energy production in the mitochondria. The difference in oxygen partial pressure between the RBCs and distal tissues is not sufficient to unload oxygen and requires the aid of "helper molecules" (i.e., allosteric effectors; see Table 1) [109]. Allosteric effectors bind to the periphery of hemoglobin subunits, outside of the heme oxygen-binding pocket. The conformational change they induce stabilizes the T-state, reducing the oxygen-binding affinity of the subunit and promoting the offloading of oxygen [115,118].

**Table 1.** Properties and allosteric effectors of vertebrate hemoglobin. 'X' marks the expression of hemoglobin properties or sensitivity to allosteric effectors. A blue 'X' represents the primary allosteric effector of the vertebrate taxa. Listed allosteric effectors include adenosine triphosphate (ATP), guanosine triphosphate (GTP), Bicarbonate ($HCO_3^-$), 2,3-Bisphosphoglycerate (2,3-BPG), inositol phosphates (IP), and chlorine ion ($Cl^-$).

| Taxon | Hemoglobin Properties | | Allosteric Effectors of Hemoglobin | | | | | |
|---|---|---|---|---|---|---|---|---|
| | Cooperativity | Bohr Effect | ATP | GTP | $HCO_3^-$ | 2,3–BPG | IP | $Cl^-$ |
| Hagfish | | | | | X | | | X |
| Lampreys | X | | X | | | | | X |
| Sharks | X | X | X | | X | X | X | X |
| Teleost Fish | X | X | **X** | **X** | X | | | X |
| Amphibians | X | X | X | | X | **X** | | X |
| Mammals | X | X | X | | X | **X** | | X |
| Turtles | X | X | X | | X | | **X** | X |
| Birds | X | X | X | | X | | **X** | X |
| Crocodilians | X | X | X | | **X** | | | X |

### 3.2. Allosteric Effector Variants

The most basal allosteric effector in vertebrate hemoglobin is ATP, which all gnathostomes share varying levels of sensitivity [108,116,119]. This effector remains the primary ligand regulating oxygen-binding affinity in basal vertebrates such as teleost fish [108,116] and is also the main ligand employed in embryonic hemoglobin across many taxa [120]. However, different tetrapod lineages possess additional distinct effectors that occupy this role. Amphibians and mammals (i.e., extant synapsids) utilize 2,3-bisphosphogylcerate (2,3-BPG) as their primary oxygen affinity regulator [108]. Conversely, in extant archosaurs (i.e., modern birds and crocodilians), the only time they possess 2,3-BPG in their RBCs is during the late stages of embryonic development [120,121], after which they become more sensitive to different organic phosphate effectors. In birds, there is a shift towards sensitivity to inositol phosphates (IPs). Compared to 2,3-BPG, IPs are more effective at reducing the

oxygen-binding affinity in hemoglobin and are therefore more efficient promoters of oxygen offloading [122]. Adult crocodilians are insensitive to both 2,3-BPG and IPs and instead are primarily sensitive to $HCO_3^-$ and acidic conditions [119,123,124]. This is an adaptation that favors their diving and ambush behavior, enabling prolonged periods underwater without the risk of muscle tissue starvation or the accumulation of lactic acid [124]. Dissolved $CO_2$ in RBCs and blood plasma interacts with water to create bicarbonate ($HCO_3^-$) and hydrogen ions ($H^+$) [70]. The $H^+$ increases the acidity of the surrounding physiological environment and binds to hemoglobin residues, further lowering oxygen-binding affinity [115,116,125], a process known as the Bohr effect [108,125]. Although birds and mammals are not as dependent on $HCO_3^-$ and the Bohr effect as crocodiles, these biochemical interactions promote the release of oxygen in distal tissues [125] in all three groups.

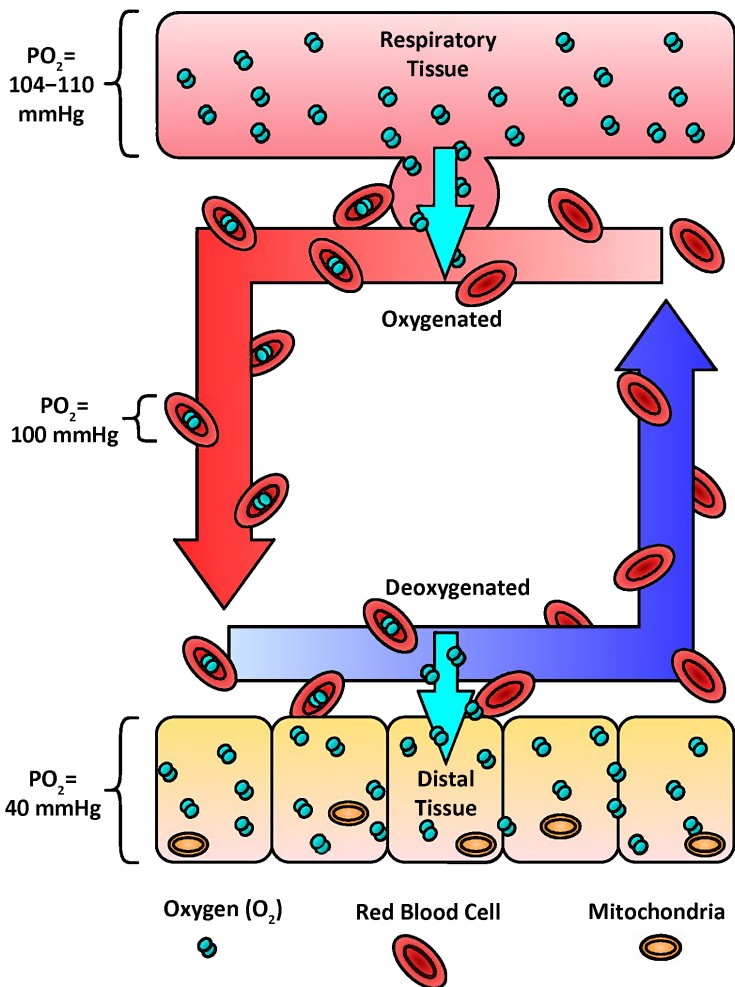

**Figure 5.** Oxygen transport. The transport of oxygen to distal tissues of the body is mediated by the difference in the partial pressure of oxygen ($PO_2$). In human lungs, the $PO_2$ ranges from 104–110 mm Hg. Oxygen diffuses down the $PO_2$ gradient from the respiratory tissue into the red blood cells (RBC) (100 mm Hg). The RBCs travel throughout the body to deliver oxygen to the mitochondria of distal tissue (40 mm Hg). Allosteric effectors support the offloading of oxygen.

### 3.3. Adaptation to Hemoglobin in Response to Extreme Environments

Because oxygen offloading is affected by factors such as environmental oxygen concentration and temperature, some vertebrates possess molecular or physiological adaptations to offset extreme environmental conditions. In the case of low oxygen environments, such as high altitudes or diving, vertebrates possess adaptations that maintain aerobic demands. One common, metabolically inexpensive strategy is to increase the production of RBCs,

thereby increasing the concentration of hemoglobin available to bind the low supply of oxygen molecules [125–127]. This has been observed in human populations indigenous to high-altitude environments, such as the Andean peoples of South America [128,129]. Another strategy is to increase hemoglobin's oxygen-binding affinity, for which both bar-headed geese and Andean camelids have hemoglobin sequence mutations [126,130,131]. Penguins, which are adapted for diving, display hemoglobin mutations that both increase their oxygen-binding affinity and additionally promote oxygen unloading via an increase in acidity (i.e., the Bohr effect) [125].

In addition to high altitude and hypoxia, extreme temperatures can affect hemoglobin's capacity for oxygen transport. Oxygen binding in hemoglobin is an exothermic (heat-releasing) process [112,132], and, as temperatures rise, hemoglobin's affinity for oxygen decreases. This effect is utilized by muscles to promote the unloading of oxygen at these tissues where exertion creates small increases in temperature [127,133–135]. However, substantial temperature increases can denature proteins, and in hemoglobin, they can greatly reduce oxygen affinity [135–137]. Some species, such as the ostrich, have developed high-temperature resistance by increasing the proportion of hydrophobic or uncharged amino acid residues in their hemoglobin sequence [138], in turn increasing their resistance to hydrolysis [139]. Conversely, in cold environments, taxa must overcome barriers to the release of oxygen to dysoxic tissue, by decreasing oxygen affinity. Therefore, cold-adapted species possess mutations that lower the energy threshold for the release of oxygen (i.e., lowering the heat of hemoglobin oxygenation; $\Delta H$) [112,113], allowing them to function over a wider range of environmental temperatures. This adaptation has been observed in reindeer, musk-ox, and reconstructed mammoth hemoglobin [112,113].

Adaptation to Hemoglobin in Transitions to Endothermy

The effect of temperature on oxygen offloading has implications for respiratory changes in lineages that transition from ectothermic to endothermic metabolisms. Ectothermy, a state in which an organism relies on environmental sources for body heat, is the ancestral trait of all vertebrates [133]. Endothermy, the self-generation of body heat, has evolved independently at various times across vertebrate taxa, most notably in mammals and birds [133]. Compared to ectotherms, endotherms have higher aerobic and metabolic demands, as well as an increased body temperature that is less dependent on the surrounding environment [127,133]. As such, the onset of endothermy necessitated greater respiratory efficiency and stability at sustained, high metabolic temperatures. This is exemplified in birds, which have transitioned from the ectothermic metabolism of their basal archosaurian ancestors to the endothermic metabolism observed in living avians [133]. The increased metabolic demand associated with flight is provided for in birds by their complex, unidirectional airflow respiratory system [133,140,141] coupled with the efficiency of IPs for their primary oxygen offloading allosteric effector. The transition between the basal ectothermic state and the derived endothermy of birds must have occurred at some point in the avian lineage, potentially in their non-avian dinosaur ancestors. The hypothesis that endothermy predates the origin of Avialae (i.e., all birds, living and extinct) is supported by the appearance of insulatory coverings (e.g., feathers) [142] and endotherm-specific behaviors (e.g., brooding) [143] in non-avian dinosaurs. Conversely, endothermy may have been present in Archosauria prior to the emergence of dinosaurs. It has been suggested that modern crocodilians are secondary ectotherms that have reverted to ectothermy from endothermic, bipedal pseudosuchian ancestors [144]. However, the timing and number of independent transitions to endothermy that occurred within Archosauria are still unclear and can only be investigated through the fossil record.

## 4. Hemes and Hemoglobin in the Fossil Record

### 4.1. Examples of Molecular Preservation, including Heme and Hemoglobin

Various studies in archeology and paleontology unveil compelling evidence of the long-term preservation of the original organic material in both relatively recent [6,15–21]

remains and deep time [10,14,22–24] remains. These studies have identified DNA [145], lipids [146], proteins [15,147], amino acids [148], and associated molecules (e.g., heme or metal cofactors) [11]. Hemoglobin has been recovered from the 5200-year-old frozen mummy, Ötzi [6,18], as well as a 10 Ka skull from *Castoroides* (i.e., giant beaver) [19]. Studies conducted on mammoth remains have revealed the presence of hemoglobin peptides among hundreds of other proteins [21,149].

The preservation of original organic compounds in deep-time fossils older than one million years old is contentious [150–152]. Conventional wisdom held that endogenous organic material could persist for less than 100,000 years for DNA or one million years for proteins [88]. However, in more recent decades, the development and utilization of molecular paleontology and paleoproteomic techniques have provided evidence of endogenous proteins across various animal taxa and geological strata. Most recently, the recovery of DNA from a 1.2 Ma mammoth has highlighted the need to reassess predictions for how long organic molecules may preserve [153].

The utilization of mass spectrometry (MS) techniques (e.g., tandem MS, or MS/MS) has recovered peptide sequences from fossil tissue over a wide age range (1 k–80 Ma) [15,21,22,24,154,155]. These include hemoglobin [19,23,149,156] and the preservation of heme cofactors [11,14] in ancient tissues. Independent of MS, immunohistology techniques (employing rigorous specificity controls to rule out cross-reactivity and bacterial contamination) [154] have demonstrated the preservation of hemoglobin in vertebrate fossil tissue [14,24,25,157–159]. As a potential proxy of endogenous hemoglobin, heme has been identified in the abdomen of a 46 Ma mosquito fossil [11], a 48 Ma fish from the Messel pit [13], a 56 Ma juvenile sea turtle [14], and in the trabecular bone of a 65 Ma *Tyrannosaurus rex* [10] (for a review, see [6]). Although hemoglobin is not the only heme-containing protein, it is unlikely these instances of heme detected in fossil tissues are derived from cytochromes, other hemeproteins, or bacteria for the following reasons: (1) there has been an observed lack of bacteria-specific peptidoglycan [14] or cytochrome [10] immunoreactivity; (2) hemoglobin is more abundant than cytochrome proteins in blood vessels [10,160] and trabecular bone [10], giving them relatively higher preservation and detection potential [161]; (3) hemoglobin demonstrates properties that inhibit bacterial growth [88,162], suggesting that tissues abundant in hemoglobin would be less susceptible to bacterial growth; (4) microbial (e.g., bacteria) structures were not observed under scanning electron microscopy [14]; (5) fossilization is likely to occur in anaerobic conditions, and anaerobic microbes exhibit a low expression of cytochrome [11]; and (6) resonance Raman Spectroscopy, which has been used to identify heme in fossil tissue [10,163,164], is capable of distinguishing hemoglobin-derived heme and cytochrome-derived heme in modern tissues [164].

### 4.2. The Preservation Potential of Heme-Based Proteins

Several factors influence the potential preservation of proteins in fossil tissue. The majority of ancient peptides are derived from preserved soft tissues (e.g., blood vessels) isolated from fossilized bones [6,19,24,155,159,165,166]. Bone and its constituent mineral, hydroxyapatite, may shield soft tissue and peptides from degradation, either by acting as a physical barrier or by providing an adsorptive surface that stabilizes proteins [161,167–169]. Proteins that are abundant within a given tissue, such as collagen in bone proteins, also have an increased likelihood of persisting across deep time [161]. Collagen in particular also benefits from a triple helical structure (i.e., tertiary structure) as the tightly complexed fibrils are resistant to enzymatic degradation [170,171]. Other factors, like the formation of cross-linkages and aggregation of peptides, enhance their survivability in deep time [5,25,88,172]. This process may be mediated by the addition of diagenetic modifiers (e.g., advanced glycation end-products, AGEs) [173] or metal cations [25,88]. Environmental and burial conditions may also influence preservation. Fossil formation favors anaerobic conditions, which reduces the impact of bacterial scavenging and growth [174,175]. Alternatively, oxidative environments may promote preservation by inducing polymerization [88,176].

Similarly, the fundamental characteristics of heme-based proteins enhance preservation potential. Schweitzer et al. (2014) proposed a chemical mechanism in which iron and oxygen derived from hemoglobin degradation increase the likelihood of soft tissue and peptide preservation [88]. Free iron and free heme released from hemoglobin along with oxygen have the potential to generate ROSs [88,177–179]. These ROS free radicals may interact with proteins and induce the formation of cross-linkages in peptides, increasing the molecule's resistance to degradation [88]. It may be even feasible that peptides attached to a cofactor (e.g., heme) may enhance peptide survivability [136].

In vivo, hemoglobin proteins degrade into individual globin chains, from which heme is separated. The globin is further broken down into amino acids and heme is converted into a noncyclic tetrapyrrole, bilirubin, by the enzyme heme oxygenase [90,178]. At death, the disruption of this enzymatic process may facilitate the incorporation of intact hemoglobin into the rock record at burial, allowing the possibility of further geochemical modification and, potentially, preservation in fossils.

Tahoun et al. review the chemical properties of heme that are hypothesized to contribute to molecular persistence in the fossil record, based on its intrinsic structure. This potential may be enhanced further by its association with hemoproteins [33]. The macrocyclic tetrapyrrole ring of the porphyrins (as discussed above) is characterized by alternating areas of high and low electron densities, which allows the molecule to dissipate energy from the environment and potentially stabilizes the molecule across deep time [29,33]. The formation of a metalloporphyrin complex with a metal cation further stabilizes the macrocyclic structure against geochemical degradation [33,43,44]. Heme and heme-like porphyrins have been described as resistant to changes in environmental pH and remain relatively unchanged at both low (e.g., pH = 0) and high pH (e.g., pH = 13.5) [180]. These molecules have also demonstrated resistance to degradation by high temperatures (330 °C) [180–182]. The hydrophobic nature of porphyrins also contributes to their stability in the rock record by increasing the molecule's resistance to hydrolysis [33,183]. In addition to their intrinsic properties, the tertiary structure and resulting resonance of hemoglobin may shield the porphyrin against enzymatic degradation and hydrolysis, increasing their likelihood of persistence in the rock record [33,183]. The chemical properties of porphyrins allow these molecules to persist across geological times and thus make porphyrins the ideal biomarker in deep-time analysis on Earth and potential applications for the search for life on other planets [184–186].

### 4.3. Degradation and Diagenesis of Heme

The heme porphyrin has been identified in organic sediment and fossil tissue samples as old as 1.1 Ga [33,187]. Tetrapyrroles (i.e., heme and chlorophylls) buried in sediment are altered by diagenetic processes into products referred to as geophorphyrins [7–9,13]. Depending on the depositional environment, these porphyrins may undergo processes such as hydrolysis, reduction, oxidation, demetallation, and remetallation [33,43]. The Treibs scheme [188] suggests that heme and chlorophylls are the biogenetic sources of the geoporphyrins etioporphyrin III and deoxophyloerythoerthoetioporphyrin (DPEP), respectively (Figure 6). It has been suggested that the diagenetic products of heme may also include the geoporphyrins di-DPEP, Rhodo-DPEP, and Rhodo-etioporphyrin, which have been described from various oils shales (Figure 7) [7–9,13,189].

Diagenetic processes that heme may potentially be subjected to, including demetallation (i.e., the removal of the metal cation) and subsequent remetallation, are of particular interest. During demetallation, the metal cation originally bound to the heme molecule is lost [43,44]. This is typically observed under acidic [43,190] and extreme heat (>300 °C) conditions [44]. In general, $Fe^{2+}$ is less susceptible to demetallation than more labile cations, such as NA, Li, and K [35]. Among the cations observed in biological systems, including those bound to proteins, iron is more stable than magnesium, and iron-bound heme is hypothesized to be more resistant to demetallation compared to the magnesium metal

cation of chlorophylls [43]. Additionally, iron cations that are oxidized (i.e., $Fe^{3+}$ when oxygen binds to $Fe^{2+}$ are more resistant to demetallation [35].

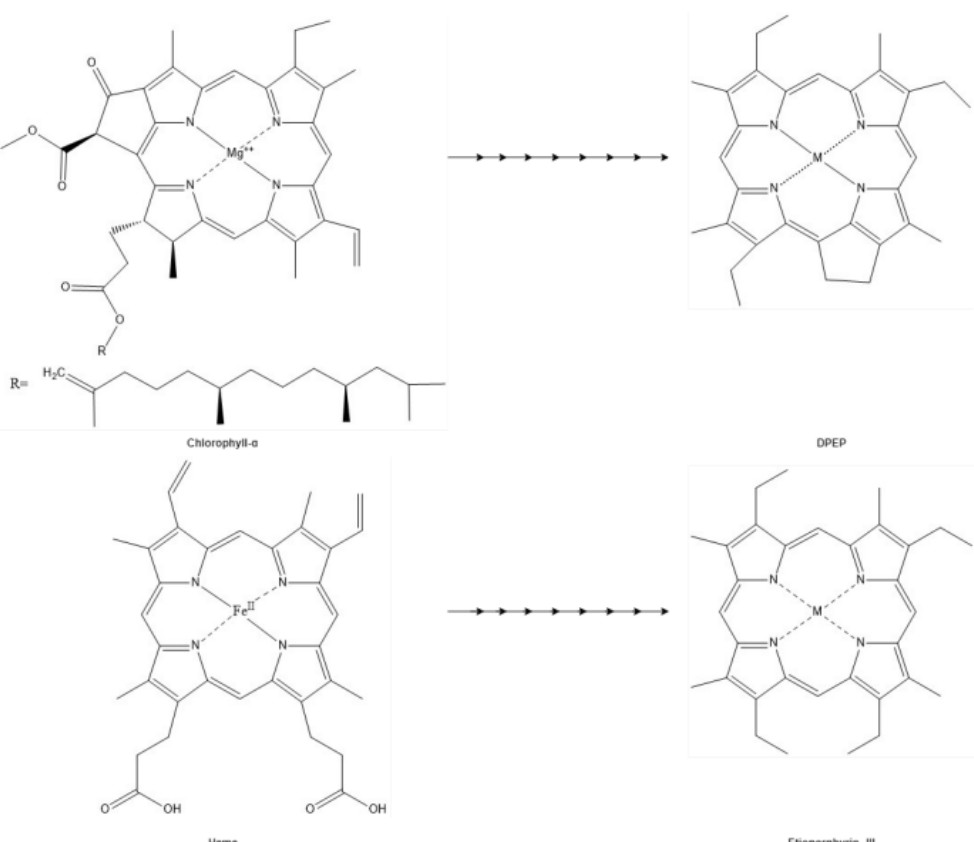

**Figure 6.** Triebs scheme. The hypothesis of DPEP and Etioporphyrin-III are geoporphyrins derived from chlorophyll and heme proposed by Triebs in 1936.

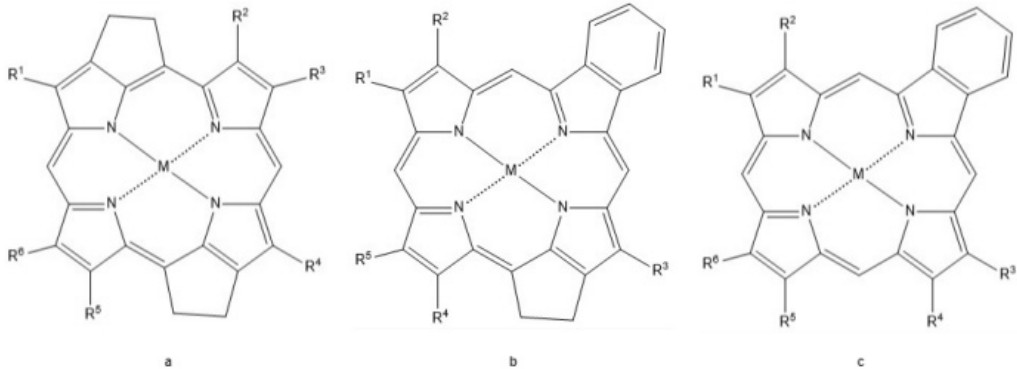

**Figure 7.** Other geoporphyrins. (**a**) Di-Deoxophyloerythoetioporphyrin (Di-DPEP), (**b**) Rhodo-DPEP, (**c**) Rhodo-etioporphyrin, porphyrins observed in the fossil records likely derived from the diagenetic alteration of heme and chlorophyll porphyrins.

Following the diagenetic removal of the original metal cations, remetallation with exogenous metal cations such as nickel (Ni) and vanadyl ($VO^{2+}$) may occur [44]. In sediment rich in organic material, nickel ions are concentrated in the reduced state ($Ni^{2+}$) [44]. Because this state is similar to the $Fe^{2+}$ ion originally bound to the heme, nickel readily occupies the vacant space at the center of the porphyrin ring [44]. The occupation of the vacant porphyrin by transition metals [44] stabilizes it [9,189]. Nickel-bound porphyrins are thermally stable and commonly occur in reduced environments [44]. $VO^{2+}$ may also chelate to heme, but this substitution is thought to occur less frequently [9,44,189]. $VO^{2+}$-

bound diagenetic porphyrins are easily reduced compared to those bound to $Ni^{2+}$ but are more resistant to oxidation [9,189,191]. In chlorophylls, the $Mg^{2+}$ of their porphyrin ring is subject to the same substitution with either nickel or vanadyl ions [9,44,189].

*4.4. Diagenesis of Globin*

The globin component of hemoglobin is itself susceptible to diagenetic alterations, beyond those described for its heme cofactor, and these may impact the preservation potential of this protein. In general, all proteins undergo a variety of diagenetic changes, producing an altered molecule called a diagenetiform [192]. These changes may include fragmentation through processes such as hydrolysis [192], deamidation of asparagine and glutamine residues [15,193], the formation of AGEs [6,15,173] or other aggregates through cross-linkages with exogenous molecules (e.g., metal ions [88,194,195]).

Despite these potential avenues of degradation, hemoglobin also possesses aspects that may increase its likelihood of persistence. For example, hemoglobin synthesis occurs within vertebrate bone marrow [37], where it may be partially shielded from the environment [176] deep below the periosteal surface. Molecular components ingrained in the structure of hemoglobin (e.g., heme, iron) have also been demonstrated to enhance preservation. For example, iron derived from heme may induce cross-linkages in hemoglobin peptides, increasing their resistance to hydrolysis and enzymatic degradation, and hemoglobin bound to oxygen has been demonstrated to enhance soft tissue preservation and possess an anti-microbial effect [88]. Because hemoglobin is a protein both specific to, and most abundant in, vertebrate RBCs [160], preserved blood vessels [165,166] and the RBC-like structures [196] found within them are an ideal target for concerted efforts at hemoglobin characterization [165,166,196].

## 5. Looking Forward: The Potential Utility of Preserved Hemoglobin

Porphyrin molecules embedded in proteins have been essential molecules for the development of life. All forms of life require the transformation and translation of energy from one form to another. During the Archean and into the Proterozoic, the earliest anaerobic metabolisms likely utilized free porphyrins, Fe-S metalloproteins, and cytochrome-like hemoproteins [55,59,69,77,180]. The evolution of oxygenic photosynthesis through the incorporation of these elements forever transformed the conditions of the planet not just in the environment but also in the metabolism of life [70,85]. As a more energetically favorable process, aerobic respiration capitalized on the increased presence of oxygen allowing for the emergence of complex life including eukaryotes and multicellular life [96,97]. With greater physical complexity aerobic organisms require a method to supplement their aerobic demands in the form of globin proteins and eventually the tetrameric hemoglobin of vertebrates [102]. The diversity of vertebrate hemoglobin is widely adaptive among taxa, allowing organisms to occupy different habitats and niches [101,108,127,132,197].

The fossil record reveals evidence of proteins, including hemoglobin, in ancient remains of vertebrates. The identification of heme-related compounds in the fossil record spans relatively recent to deep time beyond 1 Ma [6,10]. These ancient molecules have the potential to reveal information on the phylogeny, physiology, evolutionary stages, and/or processes, ecology, and burial environment of ancient organisms and ecosystems. However, to fully understand these aspects of ancient life, the process of degradation and diagenesis on proteins, peptides, and co-factors must be considered. Hemoglobin has the potential to survive diagenesis due to its protection by bone, its inherited greater concentration in RBCs, and its association with iron, heme, and oxygen.

The preservation of hemoglobin provides an invaluable opportunity to bridge the gap between the ancient past and the present. The endurance of hemoglobin into deep time allows a deeper understanding of evolutionary history and provides invaluable insights into the metabolism of extinct organisms and the influence of environmental conditions on aerobic demands.

Applying molecular paleontology techniques, including paleoproteomics and vibrational methods such as resonance Raman (RR) Spectroscopy, to well-preserved ancient tissues, including non-avian dinosaurs, holds the potential to increase our understanding of the advent and evolution of endothermy. However, the study of non-avian dinosaur hemoglobin and metabolism presents unique challenges as well, including the paucity of data for comparisons. Although non-avian dinosaurs are related to both extant birds and crocodilians [147,198], they no doubt had their own unique adaptations [133]. Birds, being endothermic, are highly active creatures adapted for flight [133], with inositol phosphate acting as their primary allosteric effector [108,130]. On the other hand, although crocodilians potentially evolved from endothermic ancestors [133,144], modern crocodiles are ectotherms. These aquatic ambush predators are known for their diving behaviors, with bicarbonate serving as their primary allosteric effector [119,124]. The divergence between these archosaur lineages makes it challenging to use phylogenetic bracketing to effectively deduce shared traits in dinosaurs. However, the analysis of fossilized hemoglobin may shed light on the metabolic processes of dinosaurs, especially maniraptoran dinosaurs, the group that eventually gave rise to modern birds [133]. In contrast, hemoglobin derived from (relatively) distantly related ornithischians may potentially demonstrate a wider diversity of dinosaur metabolism [199]. Fossil hemoglobin research holds the potential to unravel the questions that still surround the metabolic and physiological adaptations of these long-dead taxa.

**Author Contributions:** Conceptualization, J.D.A. and M.H.S.; writing—original draft preparation, J.D.A., E.R.S. and M.H.S.; writing—review and editing, J.D.A., E.R.S. and M.H.S.; visualization, J.D.A.; supervision, E.R.S. and M.H.S.; funding acquisition, M.H.S. All authors have read and agreed to the published version of the manuscript.

**Funding:** Funding for this research was provided by Lynn Orr, Susan Packard, and Vance and Gail Mullis (M.H.S.) and the Biological Sciences Department at North Carolina State University (J.D.A., E.R.S.).

**Data Availability Statement:** Not applicable.

**Acknowledgments:** The authors would like to thank W. Zheng and R. Sharma for logistical support, and two anonymous reviewers.

**Conflicts of Interest:** The authors declare no conflicts of interest.

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
