# Peer review of "Porphyrin-Based Molecules in the Fossil Record Shed Light on the Evolution of Life"

_minerals, doi:10.3390/min14020201_

Round 1
Reviewer 1 Report
Comments and Suggestions for Authors
General comments:
This is a very nicely executed review, representing very thorough research and interesting syntheses, incorporating over 190 references. Content-wise, I could not find any errors, although admittedly, I am not an expert in the more chemical matters discussed in the review. Regarding the formal aspect of the work, the text is well written and easy to read. However, there are countless little formatting errors, involving superscripts, subscripts, capitalization, etc., that could have easily been caught before submission. Since they were not, I had to mark up many of them in the annotated PDF of the manuscript. Given the extremely large number of these problem spots, I am sure I did not catch them all, so I ask the authors to do so. Another problem is the occasional repetitive style of the manuscript, where concepts, statements, and acronyms are repeated almost verbatim in several places.
Specific comments:
I only have one comment, regarding crocodile physiology: I ask the authors to amend their review by incorporating the hypothesis of crocodiles being secondary endotherms. How would that change their perspective on the allosteric effectors of oxygen transport by hemoglobin in this group?

Reviewer 2 Report
Comments and Suggestions for Authors
The author reviewed the role of porphyrins in different biological molecules and correlate their preserved structure, diversification and modification throughout the evolution. Those changes may be related and helpful in understanding the concept of evolution. In my opinion this review is great to focus some specific interaction between porphyrins and proteins in adapting evolved physiological functions.
1) How stable are the porphyrins? Is there any study on different types of porphyrins (Cu, Hg, Ru, Fe incorporated) and their stability?
2) Compared to site specific mutation, which alters the function of protein, porphyrins can cause those remarkable function differences, especially if the environment of porphyrins are same?
3) What selection pressure may be involved in changing the stable porphyrins from one form to another? Or they arose independently.
Comments on the Quality of English Languageminor typos
